# *Habitus* of Masculinity in Chilean Miners: Efficiency, Control, and Consumption of the Bodies

Jimena Silva Segovia [1,*], Paulina Salinas Meruane [2] and Estefany Castillo Ravanal [3]

1 Direction of Investigation, Universidad de Tarapacá, Arica 1000000, Chile
2 School of Journalism, Faculty of Humanities, Universidad Católica del Norte, Antofagasta 1240000, Chile
3 School of Psychology, Faculty of Humanities, Universidad Católica del Norte, Antofagasta 1240000, Chile
* Correspondence: luzsilvasegovia55@gmail.com

**Abstract:** From a gender perspective, the persistence of the *habitus* of masculinity in Chilean male miners in their family relationships, relationships with their partners, sexuality, and work was analyzed, delving into a construction that drives them toward the satisfaction of desires and consumption. The discourses of 13 workers between 25 and 62 years old were obtained through individual and group interviews. The findings constitute an axis of the research Fondecyt 1180079 carried out in Antofagasta, the region with the highest mining production in Chile and the third largest in the world. The conclusions reveal notions of bodily *habitus*, mediated by gender hierarchies, where binary masculinity is accentuated, reproducing the extractive logic of the industry in the workers themselves. Men recognize a self-demand between fulfilling the mandate of economic provider vs. a relentless disciplining of male bodies in favor of efficiency, control, and exacerbated consumption, which underlies the industry.

**Keywords:** *habitus*; body; masculinities; Chilean miners; work

## 1. Introduction

In the last 40 years, Chile has been an example of labor flexibilization, along with a subsidiary State, promoting the functioning of the market and the free flow of capital, with the main focus on the efficiency of the public administration. This model has facilitated temporary hiring, noncompliance with labor rights, and low wages in some areas of the economy, generating inequality in the labor market and high individualism, which is prone to entrepreneurship (Zabala-Villalón and Vidal-Molina 2019; Pavez and Hernández 2014) and to what Araujo and Martuccelli (2012) call a labor immoderation.

The previous issue relates to the neoliberal seal of the political, labor, economic, and social model implemented during the military dictatorship in 1973. Labor precariousness was extended through subcontracting or temporary hiring in all areas in favor of using resources and the orientation to achieve results. Thus, from free market logic, subcontracting companies have employed men and women with medium and low qualifications, perpetuating vulnerability and exclusion, with contracts without social benefits that reinforce socioeconomic and gender gaps (Valdés et al. 2014).

Then, since the 1990s, with the advent of democracy, a slow and gradual process of transformations began, not without difficulties and highly globalized, where it is evident how the economic-productive model, with some nuances, has permeated the relationships between subjects and institutions, replicating interactions of a competitive market that has generated economic growth but at the same time has deepened inequalities. According to Garretón (2012), it would be an ideology that "consists of the affirmation of the market not only as the best mechanism to allocate resources but as the model of all social or political relations, that is, as a type of society and not only of the economy" (Garretón 2012, p. 30).

Chile has gained worldwide recognition for its open economic model that facilitates international investment (see Figure 1). One of the sectors that best expresses this globalized

market is mining, with a prestigious labor supply that places it first in global exports (Segovia et al. 2021; Bravo 2012), thanks to the mineral wealth extracted mainly in the Antofagasta Region.[1] Its exploitation is 70% concentrated in the private sector at the national level, while the other 30% is in the hands of the state-owned company Corporación Nacional del Cobre de Chile, Codelco (SERNAGEOMIN 2021). At the world level, Chile has become its leading producer, with a 32% share, exceeding 5.2 million metric tons. This leadership position was gradually consolidated from 1990 onward, with the arrival of foreign investors and the increase in Codelco's capacities (Meller 2019).

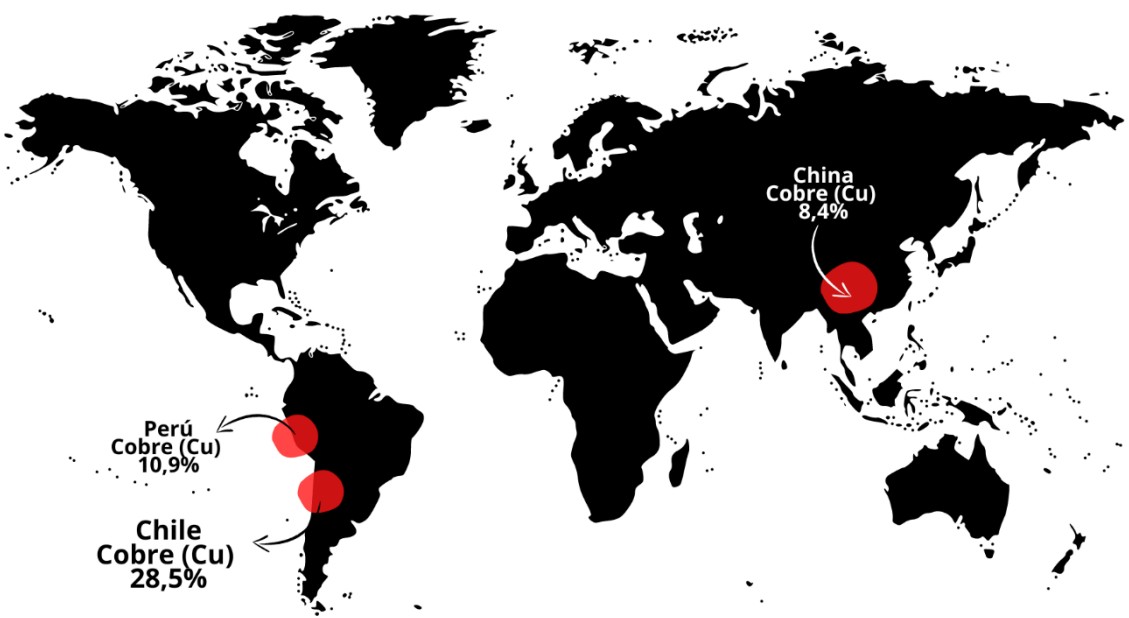

**Figure 1.** World copper production (SERNAGEOMIN 2021).

However, over the last few years, international environmental demands and the industry's high costs have begun to foster changes in production systems. Mining camps, where workers and their families lived near the deposits, were dismantled due to the harmful effects on the population's health, an enclave model dating back to the beginning of the 20th century (Zapata 1977). The "post-Fordist" transformation of the service economy was associated with the production and reproduction of the labor force, technification, productive fragmentation, the flexibilization of labor regimes, the loss of the centrality of salaried labor time in the creation of value, and subcontracting models. These changes have generated a modernizing process within companies, union relations, and the government's role (Pavez and Hernández 2014).

Although mining has positioned Chile as a world leader, a less researched counterpoint is the impact associated with the high physical and psychological demands that the organization of the industry places on workers and the consequences that this has on family dynamics. The specific interest of this study is to contribute to the understanding of the experience of Chilean men working in large-scale mining regarding the construction of masculine *habitus* in their work and affective interactions. The question is, how does the male miner (MH), under a network of masculine mandates in his work environment, construct or deconstruct his *habitus* of being a man in his body, emotions, and sex-affective interactions?

In copper-producing countries, labor dynamics have influenced society since the 20th century, fostering a culture around the deposits that have impacted the health of workers and their families. According to the findings of 16 studies reviewed[2] on pathologies developed by copper mine workers from different countries, synthesized by Cabrera-Marutz et al. (2014), they conclude that the primary occupational diseases are associated

with the environment and working conditions. In this regard, lung and respiratory tract cancer and chronic diseases such as berylliosis and silicosis stand out. The mining sector is the most affected by cancer, manufacturing, and smelting (Almonte et al. 2016).

Nehrii et al., affirm that:

> "The conditions are characterized by such factors of production environment and labor process as the work difficulty, ambient dust, illumination, noise, vibration, and microclimate. The factors may be both hazardous and dangerous for health and life of miners. Sometimes, they lead to careless and unsafe functioning practices; violations in operation sequences; mistakes by miners etc." (Nehrii et al. 2022)

Likewise, regarding mental health, a high psychological demand has been found (due to fatigue, anxiety, depression, and violence in subcontracted workers), the adverse effects of work shifts on sleep patterns, and worsening quality of life after relocation due to silica (Matamala et al. 2022).

The industry in general, and in Chile in particular, has implemented sustained changes in recent years in favor of cleaner production, such as the use of renewable energies, counteracting the high costs due to low ore deposit grades, environmental permits, innovative technological processes, and greater involvement with communities, among others (MineríaChilena 2020).

Despite these advances, there are still relevant challenges related to labor organizations and the impact this has on the physical, mental, and social health of workers. The risks associated with mining activities are not homogeneous; in contrast, their diversity is related to differences in gender, age, economic sector, branches of activity, positions held, and types of contracts, among others. For the same reason, mining exploitation has been under public scrutiny in recent years due to the negative externalities it generates, where the sacrifice zones are an example of this, and they are present in the Latin American productive model in general and in the Chilean one in particular (Hormazabal et al. 2019).

Since 2013, the application of shifts under the "7 × 7" modality (seven days worked for seven days of rest) began to spread in the national mining industry. Thus, this formula became an additional option to the existing ones, such as the 4 × 3 scheme. This trend was taken from other mining countries, as in the case of Australia, for example, where the 7 × 7 shift is widespread in Queensland (MineríaChilena 2019).

In this context, there are some differences between contracted and subcontracted workers regarding labor continuity and safety in productive tasks. Physical and psychological demands on the body mean high performance during shift work. The workers are forced to adapt their bodies to sleep–wake cycles, altitude, and temperature changes. These conditions, on the one hand, test the physical and psychological strength of the workers, who, at the same time, face the dilemma, not always sufficiently consciously, of work vs. family because the physical demands imposed by the mining industry are overlooked by the economic and social benefits offered by the industry. Faced with these conditions, the worker (with children) and his or her family group become involved and "adapt"—read sacrifice—to the mining relational dynamics, with the presence and absence of the father or mother, in the case that she is the worker and facing tensions and disconnection in intimate/emotional and interpersonal relationships with their partner and children (Segovia et al. 2016, 2021; Caamaño 2007; Klubock 1995).

The MH consulted in their relational context recognize themselves among the characteristics of masculinity as providers, protectors, self-sacrificing, active, aggressive, and demanding, especially in sexual terms, seeking to satisfy their sexual-affective desires (Connell 1995; Segovia et al. 2021).

## 2. Importance of Copper in the World

The increase in demand for mineral raw materials in the world has contributed to the increase in copper prices, which has allowed mining projects to be reactivated in countries that seemed destined for failure; this has led to the opening of new mining mills and

hiring workers with different levels of guarantees and labor rights. It has also led to strong investments in technologies, research, and development.

Regarding world demand, some aspects that should be specified in relation to copper consuming countries stand out. The irruption of China in the world economy attracts special attention, not only because of its outstanding growth, around 10% during the last years, but also because of its enormous size. Currently, China is the fourth largest economy in the world, measured by purchasing power parity, and contributed 25% of the world's growth as of 2006. For those countries in which a significant fraction of their exports corresponds to manufacturing, China has become an important competitor, as is the case of Mexico and especially Brazil in Latin America (Lehmann et al. 2007; Donoso Muñoz 2014). On the other hand, those that mainly export commodities (for example, Chile) have seen their external demand increase. The deterioration in the US housing sector, which gave rise to the global financial crisis, has influenced the demand for copper, correcting downward from 2007 onward. The community demand for copper from the European Union fell close to 4.6% in 2007 and a higher percentage during 2008, after having had a robust growth of 11% in 2006; a similar situation occurred for Japan which, from growing 4.8% in 2006, fell by 2.7% in 2007 (Donoso Muñoz 2014). These brief backgrounds allow us to understand that in these contexts large contingents of workers with different skills are required, as well as policies that guarantee quality of life and psychosocial labor well-being. In this sense, in this article, we address an axis of these experiences: *habitus*, emotions, and desires, taking into account that in different mining societies advances are being made in high-level production technologies, neglecting advances in integral well-being for the MM and his family. For this, we carry out field work in mining work areas, interviewing and talking with workers and leaders from the lower-middle sectors of the industry (see participant table), observing their work, daily, and leisure experiences. This decision responds to the fact that their working, educational, and family conditions have exposed them to greater vulnerability.

## 3. Work and Male Habitus

Work as a practice, developed early on by Durkheim (1985), is more than just a technical means for increasing the performance of the body-labor force as observed in the studied mining context. In contrast, he emphasizes it above all as a source of solidarity based on and stabilized by values such as "merit". In a type of society that he calls "primitive", this model of solidarity, paradoxically, is gestated in the structuring of a community of representations that give rise to conventions transformed into laws and that are imposed on individuals, together with uniform beliefs and practices sustained under the threat of repressive measures coming from authoritarian hierarchies, external to the solidarity group. These repressive laws constitute the foundation of what Merton (2002) called "mechanical solidarity".

The structural need for work–moral recoupling, accentuated by rationalization, autonomization, and de-normalization, remains valid even in a world where work, more than ever, is configured by an exacerbated instrumentalization in contemporary capitalism.

In mining cultures, work is historically associated with the merit of production, a resistance–sacrifice binomial that occurs between labor demands and an embodied subjectivity, which accounts for the construction of masculinity in a *habitus*. The *habitus* embodied in the worker's life and inscribed in the discourses in social relations configures an ethos that involves the worker and his family, environment, and the company. Bourdieu (2000), in the concept of *habitus*, reflects the interconnection between social structuring and individual action, which is not reduced to the application of social or individual norms but rather the introjection of the social into subjectivity, in this case, masculine.

Amuchástegui and Szasz (2007) state that the construction of masculinity is not synonymous with men but rather with a social, structural, cultural, and subjective process. It is about how male bodies embody gender practices reinforced in the social fabric. In the mining culture, workers are constructed as resistant subjects capable of controlling desires and emotions during the work shift, accentuating homosociality relations dominated by

eminently masculine languages, codes, and manners, and then readapting to the family dynamics on rest days (Barrientos et al. 2011).

Therefore, we agree with the contributions that point to the need to consider two criteria. First, to conceive masculinities as part of the everyday, relational, practical, situational, and contextual production of gender relations (Butler 2007; Connell 1995; West and Zimmerman 2009). Furthermore, masculinities are a set of creative, generative, and transformative social practices and representations (Connell 1995) set in motion by men and women but framed in specific historical possibilities and situations (Sarricolea 2017), hence their complexity and dynamism, given that in their diversity, they present features of continuity and change.

In constructing a social image of the MH in most mining cultures, sacrifice comes before the position of head of the family and, thereby, is disciplined to a set of mandates that fall on these men. This identity is a stronghold among fellow workers. It is expressed in the search for and satisfaction of desires, such as the enjoyment of consumer objects (cell phones, cars, watches, glasses, branded clothes and sneakers, various technologies for leisure), women, food, alcohol, or drugs, all status symbols in this culture, endorsed by peers, families, and even the neighborhood, underlying an illusion where exacerbated consumption is justified in the face of mining sacrifice (Barrientos et al. 2009).

## 4. Materials and Methods

This study was conducted from a sociocritical perspective to understand how the male miner (MH), under a network of masculine mandates in his work environment, organizes his *habitus* of being a man in the expressions of his body, emotions, and sex-affective interactions with women. Individual in-depth interviews and a group interview were conducted. During the analysis, through the accounts of the researched group, we identified several gender inscriptions typical of the mining culture and the construction of male subjectivity. The agreements in the private and affective life, money control in the household, and domestic organization, among others, stand out (see Table 1). By understanding these processes, we hope to contribute to the recognition of the continuities and transformations that masculinity mandates undergo in the couple's life and family ties.

**Table 1.** Example of a discourse analysis and ordering matrix.

| | | Dimension | | | | | |
|---|---|---|---|---|---|---|---|
| Categories | Subcategories | Subject involved | Subject identification | Conflicts and gender tensions | Linked emotions | Object of search | Analysis |

Author's elaboration.

The confidentiality agreement with the participants was made by employing informed consent and submitted for approval to the Ethics Committee of FONDECYT 1180079.

The selection criteria for the participants were that they should be mine workers in the Antofagasta Region, Chile, between 18 and 65 years of age, with a partner and children, and that they should work in shifts, either $7 \times 7$ or $4 \times 3$. Given the limitations imposed by the rotation system and the restrictions imposed by the companies to protect the workers' rest, with difficulty, the research team was able to gain access to the participants and conduct the interviews.

With the collected material, we analyzed the discourse of the narratives (van Dijk [2000] 2011), identifying the emerging dimensions in organized matrices, together with a brief description, as observed in Table 1. Then, we reordered them according to categories; subcategories; subjects involved in the dimension; the identification of the subjects involved; gender conflicts and tensions; emotions linked to speech; the search objective in what the subject expresses; and finally, the analysis. This organization helped move toward an in-depth and critical theoretical interpretation.

*Participants*

Thirteen men between 18 and 62 years of age participated in the study (Table 2). All mine workers were at middle levels in the command hierarchies, with partners who have been living together for at least two years. As a field of analysis, we specifically focused on the cities of Antofagasta and Calama. The selection of these spaces is justified, given the high mining production (the highest in the world), the concentration of resident workers, and the high concentration of mining leisure spaces such as shops, beer halls, and night clubs (Segovia and Delgado 2008; Segovia et al. 2021; Barrientos et al. 2009).

**Table 2.** Participants' characteristics.

| | Code | Age | Occupation | Shifts | Socioeconomic Level | Children |
|---|---|---|---|---|---|---|
| 1 | Juan | 42 | Operator-Union | 4 × 3 | Medium-Low | 2 |
| 2 | Pedro | 38 | Mine worker | 7 × 7 | Medium-Low | 2 |
| 3 | Francisco | 35 | Mine worker | 7 × 7 | Medium-Low | 3 |
| 4 | Patricio | 40 | Mine worker | 4 × 3 | Medium-Low | 1 |
| 5 | Alfonso | 45 | Operator-Union | 4 × 3 | Medium-Low | 2 |
| 6 | Manuel | 45 | Operator-Union | 4 × 3 | Medium-Low | 1 |
| 7 | Mario | 36 | Mine worker | 4 × 3 | Medium-Low | 1 |
| 8 | Julián | 41 | Mine worker | 4 × 3 | Medium-Low | 3 |
| 9 | José | 60 | Operator-Union | 4 × 3 | Medium-Low | 3 |
| 10 | César | 62 | Mechanic | 7 × 7 | Medium-Low | 5 |
| 11 | Óscar | 30 | Operator | 4 × 3 | Medium-Low | 3 |
| 12 | Roberto | 42 | Operator | 4 × 3 | Medium-Low | 1 |
| 13 | Sergio | 48 | Planner | 7 × 7 | Low | 2 |

Author's elaboration.

## 5. Discussion and Results

After the analysis of the discourses, we theorize the findings from a gender perspective (see Figure 2) that takes into account the contributions of masculinity studies by Connell (1995); Braidotti (2004); Starck and Sauer (2014); Olavarría (2001), seeking to answer how Antofagastinos male miners understand their *habitus*. Along these lines, we organize the discussion into four dimensions: (1) mining corporeality, (2) sexuality, (3) gender relations, and (4) money/power.

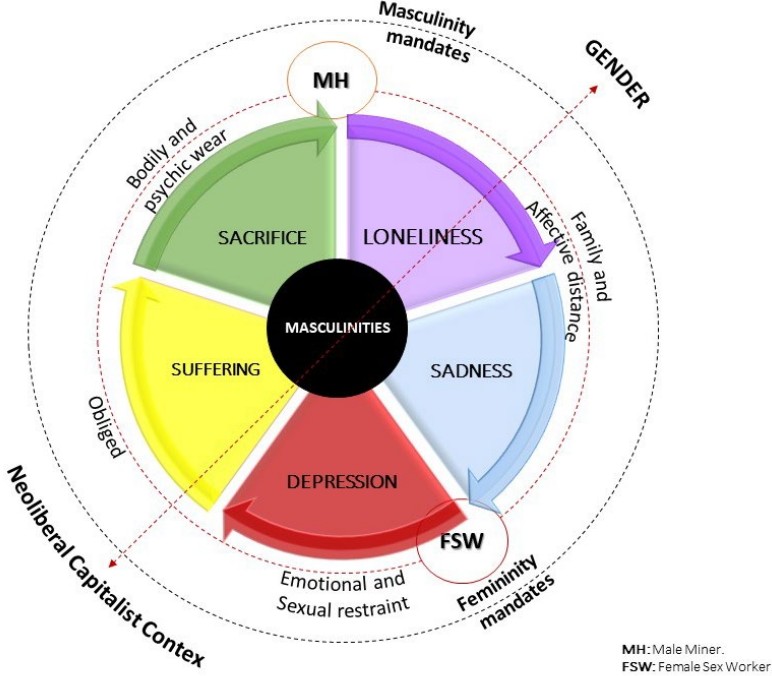

**Figure 2.** Results analysis model. Source: Author's elaboration.

*5.1. Mining Corporality*

Understanding the construction of mining masculinity implies conceiving the man–body and working as a trinomial inseparable from the hegemony of power that these subjects project in their social relations. Although it is a contradictory game between the benefits associated with this masculine status, the costs and social sanctions (implicit) fall on them when they do not comply with these mandates due to the high gender expectations involved in this mining identity (for example, in the case of extended leaves of absence due to illness or work accidents) (Connell 1995; De Martino Bermúdez 2013).

The extent of the benefits (direct or indirect) that MHs receive, compared to other workers, refers to better salaries, improved development opportunities, and the nationally recognized prestige of the industry (Segovia et al. 2021; Zuleta 2018a).

Bourdieu (1999) states that these forms of social control respond to a form of symbolic violence that operates intrapsychically. He relies on sociocultural beliefs about what it is to be a man and that, precisely in the mining industry, given its particularities, it is a problematic representation to counteract, compared to other areas, where there are nuances between patriarchal and nonpatriarchal masculinities (emerging masculinities). There is an exertion of power and pressure, including pressure on other men to reassert their virility and manhood (Zuleta 2018a, 2018b). The following story is an example of this:

> "As a man, you put up with it; a man can't solve family issues, but the mine worker is highly productive at work! I mean, the productivity of the mining worker is very high; we know that data, the old ones, solve problems quickly, are technical, look for ideas, solve, and investigate, but cannot implement those capacities at home". (Juan, 42 years old)

Once the endurance has been declared, the subject who enunciates positions himself discursively in a polarized way. In the labor scenario, "the mining worker is highly productive", an issue that in rhetorical terms leaves no room for doubt. The enunciator uses different resources to reaffirm it: he emphasizes the voice and highlights the workers' skills, but at the same time he recognizes that regarding domestic problems and family life, he feels unavoidable incompetence, a kind of disqualification to think of possible solutions. In this way, endurance is put at the service of productivity at the site; it is an object of value in the eyes of peers (other older men) up in the mine. According to Grzywacz and Marks (2000), the work and family factors influence work–family spillover differently for women in contrast to men. That is, sometimes family factors influence women's work–family spillover more for women than men, and other times men were more affected by family factors.

This masculinity is inseparable from the representation of corporeal manhood, given that it operates through demonstrations of strength and physical dexterity, implying that the minimum to be expected "from the real man is that he demonstrates that he is truly a man" (Bourdieu 2000, p. 24). Consequently, even when workers are currently more "inclined" to the discourse in favor of equal opportunities for women in the industry, the centrality of masculine attributes prevails, with primacy in their leadership and physical strength. These attributes, referring to the body and the place they occupy in the different fields of action, are built on the basis of a gender order based on a traditional and immovable male *habitus* (Silva and Méndez 2013).

Bourdieu (1991), in his analysis of the body, uses the notion of bodily hexis, later addressed by Jenkins (1993), to understand the centrality of the body in the concept of *habitus*. Hexis alludes to the behavior, manner, and style of how subjects manifest themselves in the world, behaving, speaking, or walking (Le Breton 2018). Rodríguez (2003) adds the term ethos, related to an ethical dimension that is incorporated and operates around moral dispositions anchored in the *habitus*.

> "Because if something happens to me upstairs or something happens to me because I do not control my emotions, my temperament or the conditions I do not manage, or the (emotions of) the people I am in charge of as a boss, or the

maneuvers one makes, they can lead to me either losing my job or getting hurt or hurting someone else. I get disconnected at work, which is what happens with my feelings". (Manuel, 45 years old)

This fragment expresses an idea that is reiterated in the interviews: the care of not becoming out of control at work, considering that one's own life and that of colleagues depends on the worker's ability to manage his emotions and "disconnect" from feelings. What is discursively constructed as risky goes beyond a real object of danger and encompasses that which is located in the worker's body, understood as the place of his or her emotions. Work demands emotional disconnection, a worker's bodily hexis. As Nehrii et al. 2022 mention, "mining remains the world's most dangerous industry".

In the testimonies, the mining men elaborate metaphors about their bodies and their emotions: the bodies as productive machines and their emotions as switches to connect and disconnect. For Bourdieu (1991), of all the manifestations of the person, bodily expression is the most difficult to hide since it has priority over other dimensions, given the strength of biopower, in the words of Foucault (1986). The notion of biopower reveals the regulation of bodies, considering the internalization of certain control practices deployed by the subjects. According to Weinberg (2021), mining bodies produce copper and produce themselves continuously; in addition, they forge and proudly sustain the so-called "Chilean miracle", which in recent years has initiated a gradual process of identity breakdown and questioning by the rest of Chilean society. This process has implied new negotiations in favor of the workers, commutation[3] being an example of this, although with a high impact on regional development due to the drain of resources produced by the interregional mobility of workers and their families (Stefoni et al. 2021).

Braidotti (2004) explains how the body is understood as a crossroads of the intensive forces of a subjectivized surface. Here, specific social codes are inscribed, hierarchical relations, such as geopolitical constructions marked by their position, such as boss/employee, a specific labor history, located in different scales of control such as superintendent/supervisor, boss/worker, employee/operator, hired/contractor, and others.

*5.2. Sexuality*

The concept of *habitus* developed by Bourdieu starts with the conviction of capturing reality in a structured and situated way by integrating both subjective and objective aspects. In contextual terms, the *habitus* around work is a "structuring structure" that functions as a "producer" in the MH case study. Several studies reveal how mining companies have deployed different patriarchal control mechanisms to regulate the lives of workers and their families (Weinberg 2021), prioritizing productivity and mining profitability as a priority for workers. Namely, certain principles of corporal and intersubjective actions are delivered, expressed in self-demands, family dynamics in work, repression of emotional expressions, and urgent sexual practices, among others (De Martino Bermúdez 2013; Del Carmen Rodríguez Menéndez 2003).

These practices are reinforced as *habitus* and are imposed in relationships with partners, children, or friends. For example, during rest periods, the family seeks to make up for lost time, as if the family routine could be connected or be at the service of the work cycle. The MH wants to sleep, silence, and celebrate with the sons and daughters on weekdays; they want to recover their sexual life, but the couple and the children have their daily routines. Emotional misalignments occur there.

"You want to get home to give it to the wife, and suddenly the old lady has a headache; you want to stay all day locked up in the room with her, and you can't because you have to do paperwork, you have to go out to pay bills, you have to do this and that, as they say, they give you the pass. So, with her, only the necessary because you have to do other things. In the end, nothing happens; if you're lucky, you get a bunny, the first one, a quicky, because you get to unload yourself, you don't enjoy sex, you know what I mean? That space does not exist; it becomes more and more distant. You want to give it to the wife, but she is

wearing fleece pajamas; women have been careless for a long time, making you look for distractions". (Juan, 40 years old)

The experience reinforces an image of the male sexual *habitus* of the mine worker returning from the shift: they arrive home with an urgency, a sexual drive, which transforms their desire into discharge. He wants to get to "give it to the wife" This expression refers to an idea that reproduces the logic of that *habitus* of sexual domination, in which it is he—placed in the position of the subject—who gets to "give it to the wife" placed as a relatively passive object in front of the man's action. There is a certain expectation of a woman (partner) who would expect to be sexually ready for him. However, when she arrives, this is not fulfilled because she, as the other in the relationship, exists with another desire and resists: she has a headache, is concerned about other activities, and is immersed in the domestic routine of care and in the responsibilities associated with raising the children; she is comfortable "in fleece pajamas", which acts symbolically as a barrier, discouraging seduction and desire in the eyes of the male. The worker interprets this as abandonment on the part of the woman; he says: "Women don't dress up for them". The work–shift cycle impacts the continuity of the couple's bond and the sexual-affective rhythm. They have an exacerbated desire for work isolation, and women are subsumed in the routine of child-rearing and domestic maintenance. Along this line, several studies analyze the strengthening of the patriarchal normative model, where leisure spaces oriented to MH are legitimized, such as *schoperías*, night clubs, and female sex work networks in Calama and Antofagasta (Barrientos et al. 2009; Pavez and Hernández 2014; Segovia et al. 2016).

Sex is expressed as an action that provides immediate enjoyment, supported by the purchasing and consumption power that the MH has. In these experiences, we find expressions of sex as an instance of (physical) performance and as an increasing capital, where the body is situated as an exchange commodity. Regarding the sex trade, they say:

> "She [the sex worker] has relationships with many men, and one will always be wary of having some contagion. You are looking for a woman, maybe, to go dancing for a while, have a good time, talk to her, but sex, occasionally, you pay her, and that's it. There will be many differences. Later you will find a woman you love. You want to take care of her; you want to form a family". (César, 62 years old)

In this dynamic, the sexualized other is symbolically located as an exciting and dangerous object and "relates to many"; her body is understood as something that can only be used, acquired, and consumed. In this sense, this other woman tends to be objectified and fragmented as a partial sexual object, where some parts of the body become consumable in what has been called sexual capitalism (Han [2012] 2014; Salinas Meruane et al. 2012).

*5.3. Gender Relations*

Gender is a category of analysis of power relations (Butler 2007); it reveals how distinctions develop between men and women in different fields of social interaction. These distinctions result from differentiated mandates and positions of prestige; for example, in the intimate sphere of MHs, shaping the couple's relationship, child upbringing, financial control, household organization, and others. Additionally, in production relations, there is a division of labor and its value according to gender; hence, asymmetries are distinguished: economic, labor opportunities, contractual conditions, and symbolic distinctions such as prestige (Bourdieu 2000; Connell 1995).

> "You arrive home to a house that your wife dominates, and your children no longer listen to you much; for example, you try to impose something, and in the end, the only thing they want is for you to go up to the mine again, you become a provider and no more, nothing more". (Mario, 36 years old)

> "The wife has to know how to substitute in all those moments (accidents, birthdays, parties, others); when the children are sad, the wife has to put her breast to

the kids so that they do not miss us in the time we are at work we do not miss them". (Alfonso, 42 years old)

The dichotomous gender attributions are clearly distinguished: the places men and women occupy in the mining couple. Care tasks in general, and in this case, the adequate support of children, are assigned as a woman's duty because "the missus has to", where the "has to" expresses an imperative. The imperative manifests itself as "breastfeeding the kids", which we understand as a requirement to take care of the emotional stability of boys and girls and to counteract, on the affective level, the place of the father since the father is expected "not to be absent" while he is at work. The underlying mandate is that the man's absence should not be felt. The contradiction is the ambivalence it contains: I am indispensable and dispensable, they need me, but I am not needed. The worker's link with the home and family is reduced to the money the miner provides. It is an unavoidable paradox: working for the family at the expense of the family. The need to be expendable at home takes on a nuance of reality where the benefit of the shift is imposed over the cost of absence: at home, "the only thing they want is for you to go up to the mine again".

Weinberg (2021), in the ethnography carried out in the Chuquicamata deposit (Codelco), refers to this process and explains how the copper activity, to continue growing unlimitedly, has needed to penetrate deep into the daily life of the workers and their families.

> *"During my seven days off, only one day I like to go and enjoy with my friends and my co-workers: when I go to play ball. It has to be sacred once or twice a week. However, on the other days, we try to spend with the family, enjoy and make up for those days we were absent". (Roberto, 42 years old)*

In mining, then, the relationship between the couple and the children is divided by the rhythms imposed by the work shifts. Additionally, leisure and recreational activities are filtered by the *habitus* constructed at the mine site. The MH reports enjoying a day with friends and then adds that he plays ball with them on two additional days. In summary, in three of the seven days of rest, he spends time with his friends, which he refers to as "practices of the sacred"; the subject relates this to the rituals of homosociality, where he reinforces ties with other miners. Rituals of masculinity necessary for his reputation and masculine status among peers are produced and renewed (Bourdieu 1998; Illouz 2013).

In these testimonies, we followed the line of Rodríguez (2003), complemented by Gonzalez Rey (2002). They conjugate gender *habitus* as the set of dispositions and attitudes expressed in systematic differentiation schemes between men and women. First, they mobilize the generated characteristics of a historical-cultural nature, transversalizing morals, the body, and emotions, and second, it is a system that differentiates the genders. These distinctions are embodied in the traditional division of spaces between the public and private spheres (domestic tasks), which are paradoxically recognized as complementary but are bestowed with asymmetrical power and recognition. Therefore, in the mining sector, in contrast to the greater flexibility shown in other areas of the Chilean economy (services), due to the shift system, women are (indirectly) discouraged from joining the workforce, thus perpetuating the dichotomous gender order.

Lamas (2013) states that even in androcentric organizations, women are assigned tasks of maintaining life and reproducing the labor force and the species. The women are those who perform the domestic and caregiving tasks that make it possible for men to be deployed in the labor, political, and cultural spheres, whether they prepare food and clothes for them or take care of parenting their children.

Consequently, power circulates in the nucleus of the generated *habitus* at different levels and intensities and intertwines with the couple's relationships. The prestige of work and money exercises male mining power, and at the same time, it is weakened in the intimacy of the affective bonds of the children regarding the woman/mother. Therefore, female power is unavoidable; therefore, the female *habitus* would be oriented to impose positions of domination in the household and relative "subordination" in the sexual sphere. Bourdieu (1995) argues that masculinity is constructed under the practices of eroticized

desire as a sexual relationship that seeks erotic domination. In this sense, it creates and maintains a fundamental division between the masculine and the feminine, restricting the expression of feminine desire. This distribution, being culturally constructed based on intrapsychic mandates on anatomical sexual differences, converges to sustain itself practically and metaphorically. Based on routines and repetitions, these thought patterns tend to become naturalized, making it difficult to question them, positioning men in a conflictive position of inequitable superiority over women (Lamas 2000). Masculinity is thus constructed as the residence of power, the exaltation of virility, and its association with physical or moral value (Bourdieu 1999; Zuleta 2018a).

However, being a man has mutated in the contemporary world thanks to various social, political, and economic changes (Olavarría 2017; Starck and Luyt 2019). Even from critical theories, a particular "crisis of masculinity" was promulgated; for Connell (1995), this crisis goes beyond this; the gender category has been questioned. Other authors, such as Starck and Sauer (2014), argue that it is necessary to refer to "political masculinities" whose concept should be applied to instances where power is being explicitly (re)produced or questioned, which would not be reduced only to men.

Suppose we locate the tension in the gender system. In that case, it is visible how men, on the one hand, insistently seek to maintain the cult of hegemonic masculinity. On the other hand, they support feminist reforms to deepen democracy or even to allow the freedom of expression of "new masculinities" (Kimmel 1997). For Montecino Aguirre and Acuña Moenne (1998), this tension, expressed as resistance to change in Latin American men, can be understood as a construction of masculinity based on what is not feminine, which must be demonstrated in front of others (Bourdieu 2000). Currently, there are incipient transformations in the experiences of masculinities, which invite us to avoid gender polarities and to recognize other identities that are paradoxically constructed (from language) as weak, subordinate, or sad (Olavarría 2001).

Regarding gender tensions, Olavarría (2001) states that many of the men studied in Latin America experience frustrations, discomfort, or pain as a result of their inability to respond to the demands of their work, their partners, their children, or even as a counterpoint to the hegemonic model of masculinity. Above all, because of the deterioration and wear and tear it entails, especially in the eyes of the new generations. Regarding studies carried out with Chilean men, this author states that these men do not consider their gender identities might enter into crisis with generational changes, cultural transformation, and economic and political position due to the rigidity of sociocultural hierarchies.

*5.4. Money and Masculinity*

The value of money in a capitalist economy is central to people's lives. Périlleux (2008) affirms that the leading quality of the neoliberal subject, in its genuine expression of the term, is that their disposition is to make their entire life experience an object of capitalization and mercantile negotiation. This has become an entrepreneur in itself, demanding to be active, calculating, and competitive (rather than passive and dependent), as it seems to metaphorically conceive their working life and their body as a company that must grow and bear fruit.

> "We earn more than a manager in Santiago. So, the time that I can't give to my family, I can make up for it with money. I had two girls who went to study, but did I call them on the phone? No, if they called me, fine. But [that] yes, sacred, I bought them their apartment, their car, I gave them the best education conditions I could". (Oscar, 46 years old)

"We earn more than a manager in Santiago". This assessment is based on the acquisition of money, competence, and associated benefits, which allows them to consolidate their position in a pyramidal hierarchy of positions and earnings, which, in other areas, given their training in many cases, would be unthinkable. The counterpoint, as we have already mentioned and reiterated, is the affective and emotional bonds of these MH; they are prone

to deterioration and impact the couple's relationship and position in the family, becoming a fragmented provider.

Men become vulnerable, being victims and victimizers, not constantly aware of the system's labor immoderation. Their bodies and their affections are demanded to the limit; they are in tension and internal competition with themselves and with other men and, conversely, without greater diversity among their forms of expression, many follow a continuum of mandates, without significant resistance, they have introjected duty (Connell 1995; De Martino Bermúdez 2013; Olavarría 2017). The commodification of self in work, as in sexuality, would be part of the productive subject, as it "delivers the basic conditions and limits of the organization of sexual life" (Olavarría 2017).

> "We decided that it was good that she would stay with the children for the first two years. It doesn't sound very romantic, but to this day, she is the one who is with the children, the one who carries all that burden. She is the caregiver, goes on field trips, and buys school clothes and books". (Sergio, 48 years old)

"Before, my wife worked, and we only saw each other at night. During the seven years she worked, we realized that we saw each other only three months in a row" (Mario, 36 years old).

The hierarchies based on the sex/gender system dictate the intermittent presence of the MH in the family. At the same time, the woman/mother is placed as the manager of the home and responsible for the upbringing and care of the children, leaving the father distant from these processes (Lagarde 2001). In Chile, the paradigm of the father–provider–authority family and the mother–breeder–responsible for the household is still in effect. However, with modifications resulting from the increase in female headship (39.5%), the lower responsibility of men regarding paternity, and the increase in the labor participation rate of women (52.2%), nevertheless, making work and family compatible is still a negotiation in tension, as neither at the individual nor institutional level are there sufficient conditions to facilitate the incorporation of women into the labor market (Segovia et al. 2021; Coria 1991; Olavarría 2017; Valdés et al. 1999).

Regarding the destination of money in nightlife contexts, whether *schoperías*[4] or night clubs, men compete with each other for women's attention through their financial resources (Salinas and Barrientos 2011). Women become objects of consumption that reaffirm the masculine construction (Salinas et al. 2010). This statement is found in different studies that analyze the relationship between the miner and the sex worker (Lagarde 2000; Pini and Mayes 2012; Hubbard 2004), in which it is argued that symbolic violence in societies structured under male domination constructs ideologies about women based on a binary dichotomy, where women are revered as the mother or those located in the public sphere of the night as the whore.

Before referring to sex work and its relationship with the purchasing power of the MH, it is relevant to clarify that our analysis does not pursue understanding prostitution by itself or to discuss whether it is a job or not (a problematic issue addressed extensively by Morcillo and Varela (2017); Pachajoa and Figueroa (2008); Villa Camarma (2010)). Instead, we seek to establish this practice in its relationship with men and its construction of the *habitus* of desire.

Zuleta (2018a, 2018b) argues that when a miner searches for a sex worker, he pursues another body, different from that of his partner, a body objectified at the moment, mediated by money and by the construction of masculinity *habitus*. This action allows the subject an expression of virile domination that privileges his satisfaction, reinforced by the recognition given by sex workers since they identify in their manners, languages, codes, and the hierarchy they occupy in the mining companies, either as supervisors, workers, or operators. In fact, in Calama, the *schoperias* are segmented according to these statuses; there are high-, medium-, and low-rank workers, and the aesthetics of the places, the waitresses, and the service make the difference (Barrientos et al. 2009). Han ([2012] 2014) refers to this link and explains that "the body, with its exhibition value, is equivalent to a commodity. The other is sexualized as an exciting object". Sexuality, in other words, is a consumer good in the

mining culture that is used and discarded; only its quantity is valued, acknowledging the physical endowments of the sexualized object.

## 6. Conclusions

From the research question on the network of masculine mandates of the MH in his work environment and the experiences in the *habitus* of being a man in his body, emotions, and sex-affective interactions, we conclude that the *habitus* of masculinity of the male miner operates as a system of dispositions that generate principles, practices, and representations. It is a subjective integration of moral norms (ethos) that return to the origins of ancestral mining culture.

This culture, governed by an extractivist system of natural resources and subjects, has undergone modifications in favor of cleaner production through the use of technology, renewable energies, better conditions of habilitation, reduction in shift systems, and high employability, but complex challenges remain to mitigate the intensity of self-demand, efficiency, and consumption of workers' bodies.

The force this cultural ethos has on MHs, and their families, reinforces a fiction of freedom. MHs subscribe to cultural mandates that keep them subject to the values of hegemonic masculinity; although challenged by younger generations, they remain efficient for the industry's profitability. Understanding the consolidation of power and gender tensions, that extend to the context from within the mines by companies, implies understanding, among other things, the relationship between the construction of masculinity and discipline in mining operations. The business administrations, as holders of power, try to lead, put brakes on, and control the "force of the productive masses" that generates a large number of workers coordinated in the same work process, under an androcentric and pyramidal logic. Braverman posited that "the capitalist strives, through management and planning the type of work for each worker, to control the entire process. And control is indeed the central concept of all management systems" (Braverman 1974, p. 87; Palermo and Salazar 2016). It is a control that permeates the family and its relational dynamics.

The complexity of its impact, in our understanding, is on intimate life and social relations, ultimately permeating the family's life, the affective bonds of the couple, and the orientation of desires. In this way, it shapes daily life and how people understand and respond to the world around them.

Another conclusion refers to the relevance acquired by the culture of consumption that runs through the mining ethos, determining the sense of belonging, quality, and status of these workers, reinforced at the national level in general and in the industry in particular. The money vs. family dichotomy is accentuated.

As a globalized and highly competitive industry, mining legitimizes a lifestyle in the MHs that seeks to maximize body performance, dissipate discomfort, counteract the shift system and sleep disturbances, adapt to high temperatures, and deepen a contained emotionality as part of the work experience. This rationality of life leads to the exhaustion of the physical and mental health of workers, which has been little discussed until now. Consequently, men recognize a self-demand between fulfilling the mandate of economic provider vs. a relentless discipline of male bodies in favor of efficiency, control, and the exacerbated consumption that underlies the industry.

*Contributions and Limitations*

In this study, an exploration of masculinity *habitus* was carried out in the aspects of continuity or transformation in a group of MHs from the northern Chilean zone. The field is complex and not easily accessible, and the connection achieved with mining workers' organizations is considered a valuable contribution to other male-dominated areas. We hope to contribute to the discussion and improve the working conditions of workers and their families.

We consider it important to highlight that this study is characterized by its investigative novelty, which analyzes emotions, gender *(habitus)*, and sexual-affective interactions in

the Chilean mining context. There are various investigations on mining work linked to dangerousness, work accidents, heavy work, and demands (mainly associated with men). However, the studies lack the depth of the intimate life of the MH, both of his sexual-affective relationships and of his emerging emotions and the experience of inhabiting a masculine corporality inserted in a patriarchal culture.

We believe that, for future research, there are still knots to be developed: male migration in the Antofagasta mining context; women and dissidents entering the mining world; the objectification of bodies in mining; comparison of MH experiences from different mining countries; extractivism and masculinity in the Chilean north; dynamics and distribution of tasks in couples; and the new generations of mining workers in the copper culture.

The foregoing is proposed in order to deepen the understanding of gender tensions and emotions that circulate in this androcentric culture, and together with this, open reflection on social proposals that prove the subjective well-being of society from constitutional national policies, and in expansion to international policies.

The limitations of this research are associated with the need to diversify the questions and include other aspects of mining work in the population of nonheterosexual men. It is also necessary to analyze the experience of workers' children and future professionals wishing to be part of this industry.

During the interviews, it was difficult to elucidate the experience of desires and love in the couple's relationship, the consumption of sex work, and the meanings the MHs give to these practices. These difficulties were due to the inherent constraints of a study of this nature, where the participants were not used to exploring these topics. Another obstacle was the coordination between mining shifts and the disruption of significant family time and rest periods.

**Author Contributions:** Responsible researcher, elaboration of questions, qualitative analysis, writing of a team article, J.S.S.; Writing the text, text editing, answer to the queries, P.S.M.; Co-investigator, qualitative analysis, corrections and preparation of tables, team analysis model, E.C.R. All authors have read and agreed to the published version of the manuscript.

**Funding:** This research received funding from NATIONAL FUND FOR RESEARCH AND TECHNOLOGY OF CHILE (FONDECYT 1180079). The APC for this article was funded by the researchers.

**Institutional Review Board Statement:** This research was supervised by the ethics committee of the Universidad Católica del Norte and approved by CONICYT.

**Informed Consent Statement:** Informed consent was obtained from all subjects involved in the study.

**Data Availability Statement:** The data from this research is available for consultation.

**Conflicts of Interest:** The authors declare no conflict of interest.

## Notes

1    Contributing to the 2017–2018 period, 96.5% of mining exports, meaning a total of 1,805,250 million US$ for the national GDP.

2    The countries with the most studies on the subject are the United States and Germany, followed by China, Turkey, Canada, Spain, Finland, the United Kingdom, and Sri Lanka. The complete review corresponds to 278 articles.

3    Commutation, understood as the round trip that the worker makes between his residence and his place of work.

4    Public places where beer is sold or shop, where mainly working men gather.

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
