# Peer review of "Habitus of Masculinity in Chilean Miners: Efficiency, Control, and Consumption of the Bodies"

_socsci, doi:10.3390/socsci12030119_

Round 1

Reviewer 1 Report

This article illustrates the persistence of the habitus of masculinity in Chilean male miners.

The paper is very interesting but could be improved.

There are sentences or parts of text that are not in English. For this I point out linguistic errors. Sometimes male miners are called "HM" and sometimes "MH". Only one abbreviation should be used.

INTRODUCTION:

The introduction exhaustively introduces the topic. However, in my opinion it is necessary to further problematize the research topic. From a perspective of psychological well-being, why is it so important to investigate this phenomenon on this sample with this point of view and with this methodology?

MATERIALS AND METHODS

Table 1 is incomplete or incomprehensible.

How was socioeconomic status measured?

THEORETICAL DISCUSSION AND RESULTS ANALYSIS

In figure 2 is the legend wrong? where is the abbreviation "MM"?

Participant quotes should be formatted in the same way.

Sometimes excerpts from the text are quoted in the authors' discussion. However, the literary citation is not reported. For example, the authors write in the text in italics and in quotation marks "give it to the missus" making the reader think that this part is identical to the excerpt reported. However, in the excerpt the sentence is different, in this case: "give it to the wife".

In general the discussion can be improved by emphasizing the very strong essentialist view of gender roles.

Mining Corporality: The Mining Corporality section would benefit from a mention of the spillover literature. The discourse, especially that of Juan, refer to the existence of a barrier with no work-family permeability, which prevents (positive) spillover. In this sense, a body made to work is not a body that is allowed to stay in the family by preventing spillovers, especially positive ones.

I recommend this reference and related literature.

Grzywacz, J. G., & Marks, N. F. (2000). Reconceptualizing the work–family interface: An ecological perspective on the correlates of positive and negative spillover between work and family. Journal of occupational health psychology, 5(1), 111. 

CONCLUSIONS

Directions for future research should be added and practical implications discussed.

Author Response

Dear Evaluator, we appreciate the observations and we have corrected what improves our work. Attached is a table of corrections. Thank you so much.

Reviewer 2 Report

The present manuscript describe an interesting topic. A manuscript has a practical application and also provides important theoretical for the next studies.

1. The abstract is well written.

2. The introduction is detailed but it should provide a review of the out of Chile. It will show that the raised questions are important not only for Chile but for a general society.

3. The results and discussion are well presented.

4. At the same time the tasks of the research must be highlighted at the end of the Introduction section.

5. Please provide a short description of further research.

6. There is a paper that I have reviewed in the past year*. Please consider the suggested research in your paper. I believe it worth considering in your paper.

Nehrii, S., Nehrii, T., Volkov, S., Zbykovskyy, Y., & Shvets, I. (2022). Operation complexity as one of the injury factors of coal miners. Mining of Mineral Deposits, 16(2), 95-102. https://doi.org/10.33271/mining16.02.095

Nosal, D., Konovalov, S., & Shevchenko, V. (2021). Determination of the injury probability among coal mine workers. Mining of Mineral Deposits, 15(2), 47-53. https://doi.org/10.33271/mining15.02.053

*Be aware that there are no references that belong to the reviewer

7. The novelty of the paper must be highlighted in the conclusions section.

8. In general, the presented article leaves a positive impression

Author Response

(The authors gave the same response as above.)

Round 2

Reviewer 2 Report

Dear authors,

I am more than satisfied with the corrections provided by you.

This study is an important contribution to socially sustainable mining.

Congratulations to the authors.